# Towards Sustainable Virtual Reality: Gathering Design Guidelines for Intuitive Authoring Tools

**Iolanda L. Chamusca** [1,*], **Cristiano V. Ferreira** [2], **Thiago B. Murari** [3], **Antonio L. Apolinario, Jr.** [4] **and Ingrid Winkler** [3,*]

1 Ford Motor Company, Camaçari 42831-710, Brazil
2 Technological Center of Joinville, Federal University of Santa Catarina, Rua Dona Francisca, 8300, Zona Industrial Norte, Joinville 89219-600, Brazil
3 Department of Management and Industrial Technology, SENAI CIMATEC University Center, Salvador 41650-010, Brazil
4 Department of Computer Science, Federal University of Bahia, Salvador 40170-110, Brazil
* Correspondence: ichamusc@ford.com (I.L.C.); ingrid.winkler@doc.senaicimatec.edu.br (I.W.)

**Abstract:** Virtual reality experiences are frequently created using game engines, yet they are not simple for novices and unskilled professionals who do not have programming and 3D modeling skills. Concurrently, there is a knowledge gap in software project design for intuitive virtual reality authoring tools, which were supposed to be easier to use. This study compiles design guidelines derived from a systematic literature review to contribute to the development of more intuitive virtual reality authoring tools. We searched the Scopus and Web of Science knowledge databases for studies published between 2018 and 2021 and discovered fourteen articles. We compiled fourteen requirement and feature design guidelines, such as Visual Programming, Immersive Authoring, Reutilization, Sharing and Collaboration, Metaphors, and Movement Freedom, among others. The gathered guidelines have the potential to either guide the development of new authoring tools or to evaluate the intuitiveness of existing tools. Furthermore, they can also support the development of the metaverse since virtual content creation is one of its bases.

**Keywords:** virtual reality; authoring tools; intuitiveness; digitization; sustainability; user-centered design; human–computer interaction; metaverse





## 1. Introduction

Organizations worldwide have been conducting a variety of initiatives to explore new technological solutions and to exploit their benefits, including transformations of business operations, products, processes, structures, and management concepts [1]. Digital transformation is about adopting these disruptive technological solutions, such as virtual reality (VR), to increase productivity, changing how people interact with it, either as consumers or professionals [2].

Virtual reality has helped organizations to achieve several Sustainable Development Goals (SGD). Immersive experiences improve education, raise citizen awareness, and support behavior change toward more sustainable choices, from plastic pollution to building design, as well as sustainable mobility, tourism, and water management [3], contributing to achieving SDG 5 (Gender Equality), SDG 11 (Sustainable Cities and Communities), SDG 12 (Responsible Consumption and Production), and SDG 13 (Climate Accuracy). In product development, VR has been essential to a more sustainable process, reducing carbon emissions by replacing physical products or real-world interactions with virtual ones [4], and foreseeing ergonomic risks, protecting workers from the risk of harm and improving workplace well-being [5]. As a result, it is a critical contribution to achieving SDG 8 (Decent work for all) and SDG 9 (Industry, Innovation, and Infrastructure), which calls for more sustainable industrialization, resource efficiency, and clean, environmentally

sound technology and industrial processes. Therefore, a more sustainable virtual reality contributes directly to a more sustainable future.

On the other hand, creating virtual reality experiences is not widespread and requires expensive and lengthy development processes using game engines such as Unreal (https://unity.com/pt, accessed on 7 December 2022) and Unity (https://www.unrealengine.com/en-US, accessed on 7 December 2022), which demand expert professionals [6–8]. This is due to the complexity of the software architecture of virtual reality systems, which involves a great diversity of resources including unusual input and output devices, such as head-mounted displays (HMD), tracking systems, 3D mouses, and others [9]. In addition, for a true digital transformation to take place in the economy and society we live in, it is necessary to ensure that more people are able not only to use virtual reality experiences but also to have the skill to create them.

This complex nature of immersive technology requires a multidisciplinary profile from professionals, which includes considerable technical knowledge in programming languages and/or 3D modeling [8,10–12]. Therefore, developing interactive scenes in VR is challenging and uncomfortable for novices, which include people who come from high-level creator groups, such as digital artists and designers, and developers who come from other technology fields that are not immersive [11,12]. Beginners also include professionals such as teachers, doctors, engineers, and other professionals who only want to use virtual reality as a supplement to their day-to-day work [13–15].

An alternative to the long learning curve is to adopt authoring tools, as they aim to enable efficient content creation through minimal changes. The term authoring toolrefers to software structures that include the most important tools and features of content creation while making product maintenance faster and better [16,17]. These tools are used for low-fidelity authoring, which requires less programming skills, as opposed to high-fidelity prototyping, which requires advanced programming skills [11].

There is currently a range of authoring tools for creating virtual reality experiences, with many of them available as open-source software [18]. These tools, however, are frequently limited not only in functionality but also in documentation and tutorials, making them unsuitable for supporting the entire development cycle [11,19]. This occurs because these authoring tools are often developed as proof-of-concept, to help test the user's acceptance and identify the main features they might enjoy seeing on the final product, but not with an application field in mind [16].

These factors contribute to the technology's lack of maturity, making it difficult to define best practices for multiple user applications, leading to the lack of standardized processes, the lack of recommended practices, the lack of a common language, and the lack of interoperability between virtual reality tools and between pre-existing data such as 3D assets and codes [6,11,14,17]. Combined with the complexity of virtual reality systems architecture, these barriers contribute to the persistence of novices' learning difficulties, requiring good prior knowledge to utilize the full potential of the tool [12].

Previous studies have addressed various aspects of using authoring tools for the creation of virtual reality experiences [11,16,20]. These studies complement one another, while one study introduced eight key barriers to creating virtual and augmented reality experiences [20], another investigated the challenges that virtual reality creators face when using authoring tools for collaborative teamwork [11], and another systematically reviewed studies on authoring tool development to analyze their usability evaluation methods [16].

These studies have concentrated on analyzing how authoring tools were used after they were developed and made available to users rather than on how the development process affected the final product. Therefore, there is a knowledge gap regarding the design of authoring tool software, which we seek to mitigate by listing design guidelines to assist software developers during the project definition phase. The guidelines will help them choose and create the features and requirements that these tools must have to be considered intuitive [21].

Effective human–machine communication is needed to understand probable interactions, what is happening in the present, and what can happen next [22]. The human-centered design prioritizes human needs, skills, and behaviors, and then designs to meet them. Following design guidelines [22], people can understand how to create well-designed virtual reality experiences; similarly, the guidelines of this study support the development of intuitive authoring tools to create the VR experience, but not the experience itself. The results will contribute to structuring the discussion about how beginners can accomplish their goals with a more user-friendly and inclusive authoring tool in a more sustainable VR.

The faster expansion of immersive technology can also bring a huge advantage to concepts such as the metaverse, a major digital transformation that has already been impacting the work format in the technology area, encouraging the need for vacancies for people specialized in using the metaverse in conjunction with a company's strategy. In the metaverse, users can seamlessly experience a digital life as well as make digital creations supported by the metaverse engine, particularly with the assistance of extended reality and human–computer interaction [23]. The virtual worlds that will compose the metaverse need to be created. For that, people and authoring tools will be required to enable the creation of new digital content, and the more people collaborating to build these worlds, the bigger and more diverse it will be.

Thus, the goal of the present study is to compile design guidelines derived from a systematic literature review to contribute to the development of more intuitive virtual reality authoring tools. This document is organized as follows: Section 2 describes the materials and methods utilized, Section 3 presents and analyzes the results, and Section 4 provides conclusions and suggestions for further research.

## 2. Materials and Methods

This systematic literature review adopted a qualitative approach to identify the central issues in the field, i.e., summarize the literature by pointing out the central issues [24]. The study is exploratory, which means that there has been little research on intuitive virtual reality authoring tools. This concept must be explored and comprehended, and qualitative research is particularly useful when the researcher does not know the important variables to examine [24].

This review followed the Preferred Reporting Items for Systematic Reviews and Meta-Analyses (PRISMA) guidelines, which was designed to "help systematic reviewers transparently report why the review was done, what the authors did, and what they found" [25]. Additionally, it followed a process comprising the following seven steps: planning, defining the scope, searching the published research, assessing the evidence base, synthesizing, analyzing, and writing [26]. This systematic literature review is registered on Open Science Framework, number https://osf.io/u3q7m, accessed on 7 December 2022.

As preceded by the authors of Ref. [4], an expert in virtual reality-based design testing of product development defined the initial search strategy, which was then assessed individually by three senior researchers. Qualitative research is interpretative research [24]; therefore, it is relevant to the outcomes of this study that the authors have a strong background and experience with 3D visualization tools, graphical design, computer-aided design, and software such as SolidWorks, Adobe Photoshop, CATIA, and Autodesk, among others.

The strategy resulting from this validation process is described in the sections that follow.

### 2.1. Planning

The knowledge bases that will be investigated are determined during the planning step [26]. The investigation was carried out using the scientific databases Scopus and Web of Science. These databases were chosen because they are reliable, multi-disciplinary scientific databases of international scope with comprehensive coverage of citation indexing,

providing the best data from scientific publications. Scopus now includes 87 million curated documents [27], whereas the Web of Science covers more than 82 million entries [28].

*2.2. Defining the Scope*

Defining the scope means presenting proper research questions; therefore, three main questions were selected for this systematic review:

Q1: What are the characteristics of the virtual reality authoring tools reviewed? Q2: What is the definition of intuitiveness in virtual reality authoring tools? Q3: What are the guidelines for designing intuitive virtual reality authoring tools?

*2.3. Literature Search*

In the literature search step, a particular string is used to search the database set up in the planning step, based on the research questions asked in the defining the scope step [26]. The first keywords utilized were *virtual reality* and *VR* to specify the search for this specific cut from immersive technology; therefore, not considering augmented reality, with the goal of defining, at the end of this study, design guidelines more focused on virtual reality experiences. The exploratory search was based on articles focused on describing the process to develop *authoring tools*, specifically oriented to create virtual reality experiences, followed by complementary keywords, *system* and *frameworks*, added to the string to cover other types of software tools.

Because virtual reality experiences can be difficult for novice users to create, the focus is on keywords linked to intuitiveness. From the word *intuitive*, other synonyms were gathered: *flexible*, *democratize*, *adaptable*, *usable*, *facilitate*, *simplify*, *easy* and *user-friendly*. Thus was formed the final search phrase: TITLE-ABS ((*virtual reality* OR *VR*) AND (*authoring tools* OR *system* OR *framework* AND (*intuitive* OR *flexible* OR *democratize* OR *adaptable* OR *usable* OR *facilitate* OR *simplify* OR *easy* OR *user-friendly*)).

*2.4. Assessing the Evidence Base*

The assessing step uses inclusion and exclusion criteria filters to reduce the number of documents found in the searching the literature step—selecting those that are relevant to the research questions [26]. These criteria were applied to the researched articles in three phases, as follows:

Phase 1: exclusions through filter options provided by the database used in the research.

- E1.1.: The entry title or abstract did not have one or more of the terms described in the search phrase;
- E1.2.: Published before 2018;
- E1.3.: Entry not written in the English language;
- E1.4.: Virtual reality is not a keyword;
- E1.5.: Duplicate entry.

Phase 2: exclusions through screening of the abstract of publications.

- E2.1.: Entry is theoretical work (e.g., information system proposal, literature review, poster);
- E2.2.: Entry does not consider the development of authoring tools for virtual reality immersive experiences creation;
- E2.3.: Entry focus on augmented reality;
- E2.4.: Entry develops authoring tools for virtual reality experience creation not based on the use of HMD on virtual environments (e.g., CAVE, 360 video).

Phase 3: exclusion through screening of the entire article, using the tool Mendeley Desktop for organizing and classifying the publications.

- E3.1.: Entry with less than 5 pages;
- E3.2.: Entry related to the development of authoring tools not directly defined as intuitive and easy to use for beginners and unskilled professionals;

- E3.3.: Entry limited on the development of authoring tools specific to an area of application (e.g., health, engineering, education, and culture).

The research period chosen was between 2018 and 2021, the last four years before the development of this study; a period equivalent to the fast progression of this technology through these years, not only academically but also in virtual reality hardware releases. For example, the advent of technologically advanced virtual reality headsets in 2016 represented a breakthrough for virtual reality applications and practitioners with the release of the HTC Vive Steam VR headset, the first commercial release of sensor-based tracking [29,30]. Another significant event in the evolution of virtual reality headsets occurred in 2018, with Oculus launched the Oculus Go, the first commercially available wireless virtual reality headset with an affordable built-in screen [31].

### 2.5. Synthesizing and Analyzing

Figure 1 depicts the flow of the systematic review from searching the published research to synthesizing processes.

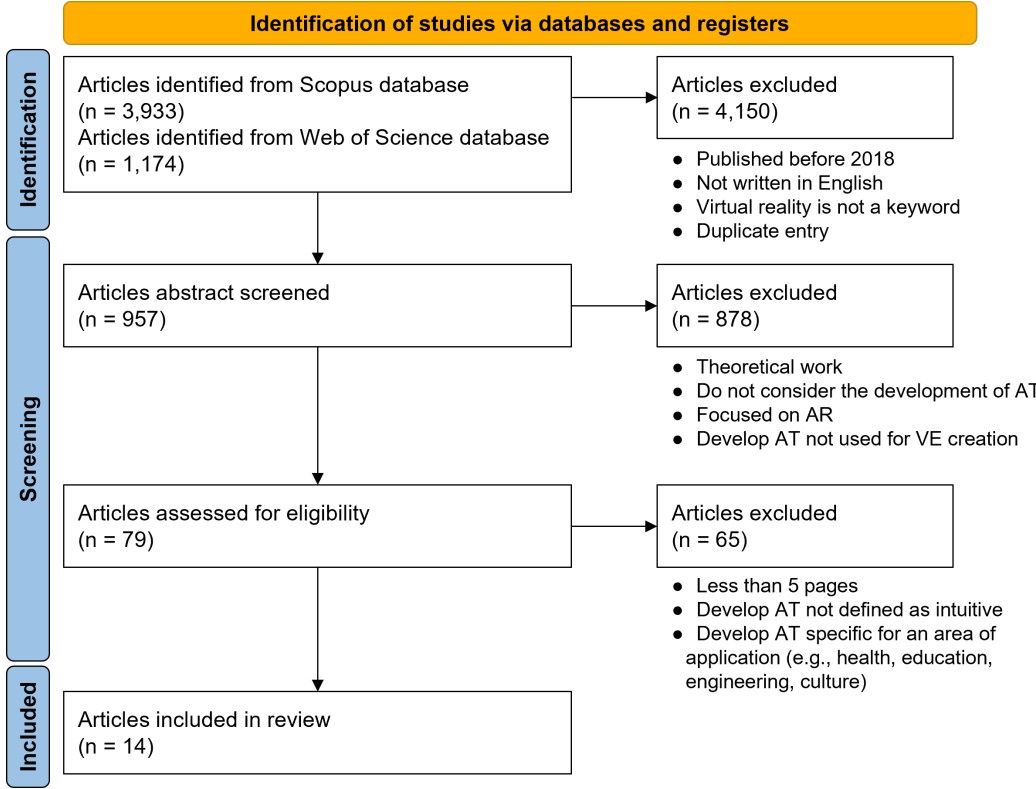

**Figure 1.** Systematic review flow diagram, adapted from PRISMA [25].

Textual information is analyzed in qualitative research designs, allowing researchers to interpret themes or patterns that emerge from the data [24]. The procedures for data analysis aim to extract meaning from the text; they entail segmenting, deconstructing, and reconstructing the data [24]. In terms of data analysis steps and procedures used for interpreting and validating collected data, we employed text non-numerical analysis, with the type of interpretation consisting of themes and patterns to identify design guidelines, and peer experts debriefing strategy for validating findings.

Regarding the non-numeric analysis and interpretation of themes and patterns, we began by reading the retrieved articles through Mendeley Desktop and describing the characteristics of virtual reality authoring tools (Table 1) in terms of which *artifact* was developed in the study, the *software* and *hardware* used and *plugin* or *standalone* type, where a plugin is "software developed to work over other software to facilitate processes" and a

standalone is "a software that works without any other software and is designed specifically for a purpose" [16].

**Table 1.** Characteristics of the virtual reality authoring tools.

| Ref | Artifact | Hardware | Software | Type |
|---|---|---|---|---|
| [32] | Tool for 3D assets search through immersive handmaid sketch in VR | Oculus Consumer Version 1 (HMD) and Oculus Touch Controllers | Unity3D Engine and Convolutional Neural Network (CNN) | Plugin |
| [33] | Tool for visual feedback of haptic properties in VR | HTC Vive Kit (HMD) | Unity3D Engine | Plugin |
| [34] | Collaborative web-based authoring tool for creating virtual environments in VR | Oculus Rift Consumer version and HTC Vive (HMD) | Node.js/EasyRTC, Socket.IO, WebRTC and A-FRAME (Three.js + WebVR) | Standalone |
| [35] | Architecture for a collaborative immersive tool for multisensory experiences creation in VR | HTC Vive (HMD), Bose QuietComfort 25 headphones, Sensory Co SmX-4D for Smell, Buttkickers LFE kit and wind simulator | Unity3D Engine | Plugin |
| [36] | Immersive authoring tool that allows applying reaction behaviors to objects using visual programming | Oculus Rift (HMD) | NOT INFORMED | Standalone |
| [37] | Tool for 3D assets editing and application of behaviors to objects | HTC Vive Pro Eye (HMD) and controllers | Unity3D Engine (C#) and SteamVR | Plugin |
| [38] | Authoring tool for developing interactive agents for VR applications | NOT INFORMED | Unity3D Engine and Cortana for speech recognition | Plugin |
| [8] | Immersive authoring tool that allows applying reaction behaviors to objects using visual programming | NOT INFORMED | A-FRAME (Three.js and WebVR), Node.js and RxJS | Standalone |
| [17] | Immersive authoring tool with visual programming to reproduce gamified training scenarios through modular architecture | Oculus Quest 2 (HMD), HTC Vive (HMD), Microsoft Hololens (HMD) and Magic Leap | Unity3D Engine and CodeDOM (.NET Framework) | Plugin |
| [39] | Authoring tool that uses "nugget tiles" (blocks) so authors can create reduced learning experiences in VR | HTC Vive (HMD) | Unity3D Engine and Virtual Reality Tool-kit (VRTK) | Plugin |
| [10] | Platform for interactive virtual objects creation in VR through physical objects presented in the real world, using AI | Oculus Quest (HMD), Oculus Link, Touch controllers and iPad Pro | Unity3D Engine and AppScanner | Plugin |
| [40] | Authoring tool that allows flexible control of work spaces for data analysis during collaborative activities in groups inside an immersive space in VR | HTC Vive Pro (HMD), Backpack | Unity3D Engine, Immersive Analytics Toolkit (IATK) and Oculus Avatar SDK (body) | Plugin |
| [41] | Web-based authoring tool with better graphic quality | NOT INFORMED | Unity3D Engine and WordPress/MySQL | Plugin |
| [12] | Web-based intuitive authoring tool for interactive 3D scenes creation in VR | NOT INFORMED | A-FRAME (Three.js), React and JavaScript | Standalone |

Then, using Microsoft Excel, we hand-coded the data collected in the studies to generate the design guidelines, i.e., we arranged the text data gathered by bracketing segments into categories and labeled them with a title. The coding process was then utilized to provide a description of the categories for analysis, which are the themes that often arise as major findings in qualitative investigations.

Based on the authors' interpretation, a method inspired by the agile methodologies artifacts [42] was used. The product backlog, one of the artifacts produced by the scrum process, is a prioritized list of product requirements or features that provide business value for the customer [21]. In this study, the virtual reality authoring tools were considered as *products*, and the non-expert users as *customers*. Therefore, the identified design guidelines were also grouped into *requirements* (generic concepts that delimit the major characteristics of a product by understanding the user's needs before the development starts and will help define the product's features), or *features* (tools one uses within a system to complete a set of tasks or actions, the functionality of a feature provide the user a desired outcome).

Finally, we discussed each identified design guideline in depth in a qualitative narrative, along with quotations from the reviewed works from which we interpreted that guideline, to support the categorized themes [24]. Although we recognized numerous quotations for each identified design guideline, we highlighted just three quotations as illustrative instances of each design guideline for the purpose of this article's writing flow; the entire list of quotations is accessible as Supplementary Materials. Using the authors' qualitative understanding, a list of *related terms* for each guideline was also compiled, which is primarily made up of synonyms. However, these terms are not connected to the frequency with which they appear in the reviewed articles, since having a higher frequency of appearance in the articles does not make them more important in the context of our analysis, as well as all the guidelines have the same weight in terms of relevance. The information flow summarizing all the steps covered in this section is presented in Figure 2.

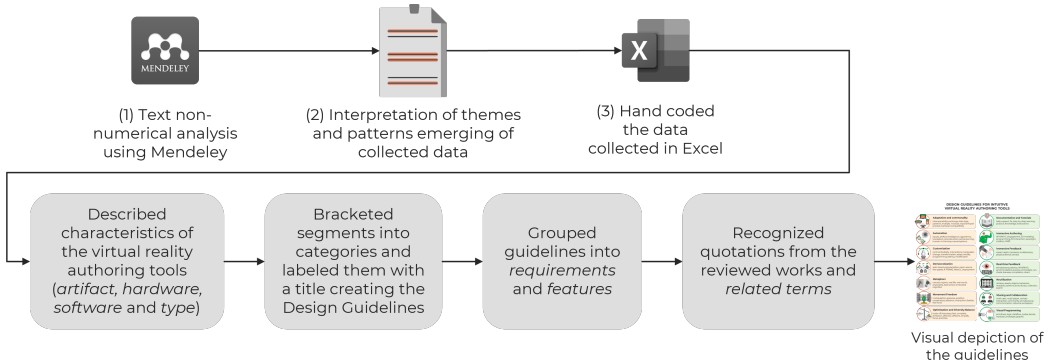

**Figure 2.** Synthesizing flow diagram.

Many qualitative studies include visuals, figures, or tables as adjuncts to the discussions [24], so we generated a visual depiction of the developed design guidelines, as shown in Section 3).

Qualitative validity means that the researcher checks for the accuracy of the findings by employing certain procedures; is based on determining whether the findings are accurate from the standpoint of the researcher or the readers of an account [24]. We used the peer debriefing strategy to assess the validity of the identified design guidelines, which involved peer debriefs reviewing and asking questions about the qualitative study at field conferences such as the IEEE International Symposium on Mixed and Augmented Reality (ISMAR) so that the account would resonate with researchers other than the authors. This strategy, which involves an interpretation beyond the authors' view, adds validity to an account.

This process led to the fourteen design guidelines presented in the following section.

## 3. Results

In the following sections, the research questions Q1, Q2, and Q3 are addressed.

### 3.1. Characteristics of the Virtual Reality Authoring Tools

The fourteen studies were reviewed to analyze the research question *Q1: What are the characteristics of the virtual reality authoring tools reviewed?*, and the results are presented in this section. The following is a summary of the main characteristics of the virtual reality authoring tools developed by the reviewed studies:

1.  Virtual environment creation, where everything that the user sees is a 3D model, also containing collaborative interaction, visual programming, and immersive authoring [16];
2.  Generic purpose, not developed for the use in a specific field, such as Mechanical Engineering or Medicine [16];
3.  Manipulating and importing 3D objects by searching online, either by text or with an immersive sketch in VR mode, editing assets, and adding behaviors;
4.  Facilitating interaction between software and hardware through haptic feedback visualization and multisensory stimuli;
5.  Interactive human characters development, giving the user pre-setted behaviors such as mouth movements to speak;
6.  Artificial intelligence automation using different types of networks to help the user achieve their goals with more efficiency.

Item 1 also represents an exclusion criterion described in Section 2.4, which was defined by the fact that most of the authoring tools developed in the field today are aimed at creating 360° videos [16]. The 360° experiences allow users to look around freely and are very simple to create, only requiring the application of one or more video files in a virtual reality context to work, nonetheless valuable information can be lost. All the potential interactivity with elements in virtual reality is wasted in a 360° experience, which is why it is a major limitation that the authoring tool only supports 360° videos. In addition, as the authoring tools for creating virtual environments are more complex to use, the present study will have greater value in its contribution towards the intuitiveness of creating VR experiences in virtual environments.

Examining Table 1, the *standalone* authoring tools on the reviewed works were always the web-based ones, using A-FRAME to be built, which is easier to use for proof-of-concept platforms, while the *plugin* authoring tools were always based on Unity3D Engine, a mainstream non-paid game engine. This shows that the authoring tools (standalone and plugins) made in the reviewed works are not ready for the market yet because they have not been released as final products. Table 1 summarizes the variables described in each article:

### 3.2. Definition of Intuitiveness in Virtual Reality Authoring Tools

The fourteen studies were examined in regard to the research question *Q2: What is the definition of intuitiveness in virtual reality authoring tools?*. The reviewed articles highlight intuitiveness as *easy-to-use*, *quickly*, *high usability*, *for non-experts*, *short training*, *simple*, *facilitate* and *reduce complexity*. In these studies, intuitiveness is related to completing tasks quickly, requiring minimal learning, lowering the entry barrier, reducing information, time, and steps, being appropriate for both expert and non-expert users, being aware of and feeling present in virtual reality, feeling comfortable with the tool, making few mistakes, and using natural movements in virtual reality.

It is not possible to evaluate or measure intuitiveness, but we may measure it with usability, effectiveness, efficiency, and satisfaction, using well-established questionnaires and methods (some of them listed in Table 2). For example, usability can be measured with System Usability Scale (SUS), effectiveness can be measured by tasks completed successfully, the number of errors, and the number of help requests, efficiency can be measured by time spent to complete a task and perceived workload, and satisfaction can be measured with the After-Scenario Questionnaire (ASQ) [16,43]. Other measures can

be taken into account, such as learnability (time to learn a tool) and recommendations to others.

High-technology products need to exhibit good usability, a qualitative measure of the ease with which a human can employ the functions and features offered [21]. In this study, *intuitive* refers to the quality of an easy-to-use authoring tool whose usability, effectiveness, efficiency, and satisfaction evaluation showed positive results.

In terms of usability evaluation methods, nine studies adopted Likert-scale surveys [8,10,12,17,32,33,37,40,41], three used SUS [10,12,35], four adopted other types of questionnaires such as ASQ and NASA TLX [35,37,39,41], three used qualitative retrospective interviews [8,10,36], two implemented other methods such as the thinking-aloud method or measured the number of errors and time taken to complete the activity [38,39], and only one did not use any evaluation methods [34]. Table 2 shows how almost all of the authoring tools, using various methods of evaluation, presented similar conclusions, which are often described in terms of being intuitive.

**Table 2.** Detecting *intuitiveness* in the virtual reality authoring tools.

| Ref. | Intuitiveness Quote |
|---|---|
| [32] | "[…] users can perform search more quickly and intuitively […]" |
| [33] | "[…] rapidly create haptic feedback after a short training session." |
| [34] | "The tool features an intuitive and easy to use graphical user interface appropriate for non-expert users." |
| [35] | "[…] positive feedback from users regarding ease of use and acceptability." "[…] an authoring tool that is intuitive and easy to use." |
| [36] | "[…] most participants commented positively on this application and […] expressed that the application is easier for beginners." |
| [37] | "AffordIt! offers an intuitive solution […], show high usability with low workload ratings." |
| [38] | "[…] people with little or even no experience […] can install VAIF and build interaction scenes from scratch, with relatively low rates of encountering problem episodes." |
| [8] | "FlowMatic introduces […] intuitive interactions, […], reducing complexity, and programmatically creating/destroying objects in a scene." |
| [17] | "[…] efficient data structure, for simple creation, easy maintenance and fast traversal […] users can create VR training scenarios without advanced programming knowledge." |
| [39] | "[…] An immersive nugget-based approach is facilitating the authoring of VR learning content for laymen authors." |
| [10] | "The interaction procedures are simple, easy to understand and use, and don't demand any specific skill expertise from users." |
| [40] | "We choose user interface elements […] to minimize learning time […]" "it was useful to see each others' work in real time to improve workspace awareness, and it was easy to share findings with one another." |
| [41] | "Evaluation results indicate the positive adoption of non-experts in programming […] participants felt somewhat comfortable using the system, considering it also as simple to use." |
| [12] | "[…] we have analyzed the effectiveness, efficiency, and user satisfaction of VREUD which shows promising results to empower novices in creating their interactive VR scenes." |

### 3.3. Defining the Design Guidelines

The fourteen studies were examined in regard to the research question *Q3: What are the guidelines for designing intuitive virtual reality authoring tools?* and the results are presented in the following section.

Getting the proper virtual reality specifications is difficult, and a creator's first few projects often fail [22]. Well-designed virtual reality experiences may increase performance and save costs, give new worlds to explore, boost education, and develop deeper comprehension by letting users walk in someone else's shoes. However, even virtual reality professionals cannot always effectively define a new project from the start since good virtual reality design combines technology and human perception. Trying to mitigate this

challenge, design guidelines help authors to create better virtual reality experiences [22], which is the same strategy we use in the present study but in a different context. As well as Pressman [21] design guidelines were gathered to help software developers, specifically aiming to support the development of intuitive virtual reality authoring tools.

Thereafter, the design guidelines will help beginner authors break the barrier of starting to create in virtual reality. Moreover, since most of the authoring tools found in the reviewed works are only proof-of-concept, the guidelines can encourage the development of mainstream platforms with fewer limitations, democratizing the technology and increasing its maturity. Figure 3 shows how software developers can use design guidelines during the development process of authoring tools:

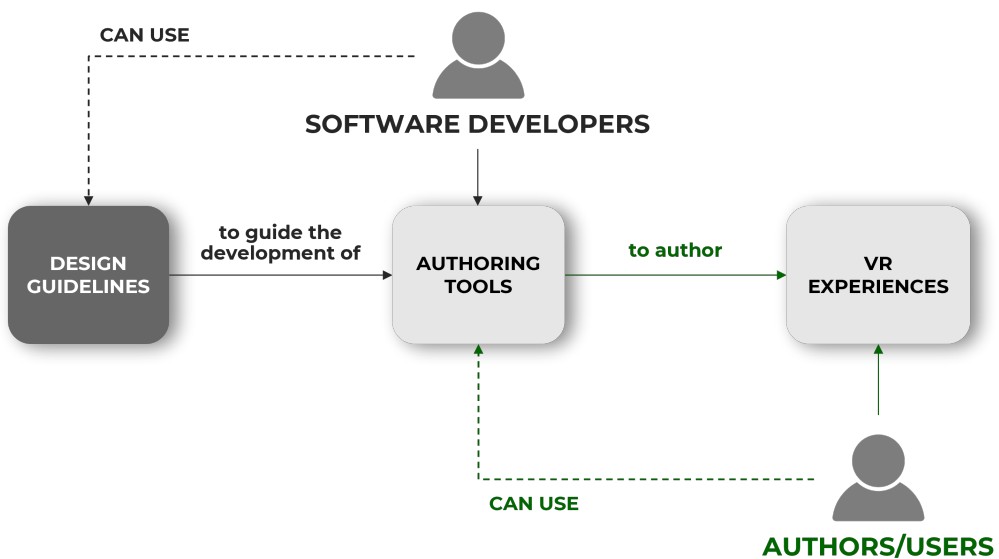

**Figure 3.** Information flow of the design guideline application in the authoring tool life cycle.

The lack of ontologies related to the concepts of virtual reality authoring tools is examined [16], indicating that there are few connected standards for the development of these platforms. In other words, concepts, methods, and nomenclature are not well-established, resulting in the development of authoring tools with vastly different formats and the application of diverse evaluation techniques to determine their usability. Similarly, the need for a taxonomy proposal for the metaverse was addressed since the wide scope of this concept causes a lack of understanding about how it works [44]. Between the proposed taxonomies, we can point out the *components* that are thought to be necessary for the realization of the metaverse, namely hardware, software, and contents. Many similarities were found between the design guidelines suggested in the current study and technologies that have recently become issues and interests to the metaverse and were mapped as hardware, software, and content [44]. This adds to the belief that the guidelines can positively contribute to the creation of the metaverse, through their influence in facilitating the use of the components that form them.

Moreover, our findings also contribute to advancing the creation of ontologies for the development of virtual reality authoring tools in relation to the current gaps [16]. Due to the lack of ontologies for authoring tools, the concepts, and common functions among the authoring tools analyzed, often used different terms to refer to the same element. It is important to highlight that the guidelines obtained complement each other, and were never presented in isolation. The non-identification of a guideline in a given work does not mean that the authoring tool does not use it, it only means that it was not mentioned in the article description. Moreover, the identification of a guideline in the article is not necessarily linked to its presence in the tool, but it may have been cited as an application of

previous work or an intention to improve the tool in the future. Table 3 summarizes the design guidelines and their application.

**Table 3.** Design Guidelines list, classification, articles, and related terms.

| N | Design Guidelines | Classif. | Articles | Related Terms |
|---|---|---|---|---|
| 1 | Adaptation and commonality | Requirement | [8,12,17,34,35,38–41] | interoperability, exchange, data type, patterns, multiple, modular, export/import process, hardware compatibility |
| 2 | Automation | | [10,12,17,32,37,38,41] | inputs, artificial intelligence, algorithms, translation, reconstruction, active learning, human-in-the-loop, neural systems |
| 3 | Customization | | [8,10,12,34–41] | control, flexibility, interactions, manipulate, change, transformation, adapt, modify, programming, editing, modification |
| 4 | Democratization | | [8,12,34,38–41] | web-based, popularization, open-source, free assets, A-FRAME, WebGL, deployment |
| 5 | Metaphors | | [8,10,12,17,32–34,36,37,39,40] | natural, organic, real life, real-world, physicality, abstraction; embodied cognition |
| 6 | Movement Freedom | | [8,10,32,34,36,37,39,40] | manipulation, gestures, position, unrestricted, selection, interaction, flexible, free-form |
| 7 | Optimization and Diversity Balance | | [8,10,12,17,32,35,37,39–41] | trade-off, less steps, fast, complete, limitation, effective, efficient, simplify, focus, priorities |
| 8 | Documentation and Tutorials | Feature | [8,12,17,37,38,41] | help, support, fix, step-by-step, learning, practice, knowledge, instructions |
| 9 | Immersive Authoring | | [8,10,12,17,32,34–37,39,40] | WYSIWYG, engagement, 3D modeling, programming, 3D interaction, paradigm, creation, HMD |
| 10 | Immersive Feedback | | [33,35–37] | visual, haptic, hardware, multisensory, physical stimuli, senses |
| 11 | Real-time Feedback | | [8,12,17,32–37,39,40] | simultaneous, latency, WYSIWYG, synchronization, preview, immediate, run-mode, liveness, compilation, direct |
| 12 | Reutilization | | [8,10,12,17,32,34,36,38,39,41] | retrieve, assets, objects, behaviors, reusable, patterns, store, library, collection, search |
| 13 | Sharing and Collaboration | | [12,34,35,40,41] | multi-user, multi-player, remote interaction, community, simultaneous, communication, network, workspace |
| 14 | Visual Programming | | [8,17,36,39,41] | primitives, logic, dataflow, nodes, blocks, modular, prototype, graphic |

The next sections describe the design guidelines classified as requirements.

### 3.3.1. Adaptation and Commonality

This guideline relates to *interoperability* [44], to enable the integration of different data acquisition sources (hardware or software), adaptable to a wide variety of cases and purposes; being usable for any application field (education, science, history, business, culture, and design); allowing communication with different types of virtual reality hardware, such as different head-mounted displays, controllers, and wearables such as haptic gloves and clothes; use of patterns, blocks, nodes, or modules to organize functionalities; allowing content creators to set up different modules by turning on and off plugins, which can also change the tool's user interface; use of the same tool to create for different platforms, such as personal computers, head-mounted displays, and mobile smartphones; use of common programming languages and/or have the ability to use a known language of the user's choice; acceptance of different file extensions for the same type of data, such as .fbx, .catpart, and .igs, which are all extensions for 3D data from different types of 3D modeling; having a unique file extension that could retain all kinds of information and be used by all software would be the perfect scenario, which is performed by Universal Scene Description (USD) files, an extension created by Pixar. The following examples illustrate the guideline:

- "To ensure a proper multisensory delivery, the authoring tool must communicate effectively with the output devices" [35];
- "using semantically data from heterogeneous resources" [17];
- "Establishing an exchange format and standardizing the concept of VR nuggets is a next step that can help to make it accessible for a greater community" [39].

### 3.3.2. Automation

This guideline concerns automatic processing of activities that would require human interference, where algorithms must complement the human creative work to avoid non-productive activities; the use of an artificial intelligence network such as CNN or GAN to create systems that can analyze inputs and come up with better results; the use of simple sketch drawings to search for equivalent 3D models; scanning the physical world through complementary hardware such as LiDARs or smartphones and use a raw point cloud to retrieve better virtual models; production of 3D models out of 2D images; provide autonomous tools to segment the 3D mesh into minor parts; triangle reduction on high polygon objects; follow human repetitive activities to create codes to reproduce them (human-in-the-loop); prediction of actions with smart suggestions, such as adding functions and behaviors to objects; artificial intelligence assistant to provide tutorials and help as needed; use various inputs, such as voice commands, to activate a functionality [44]; translation into different languages; using cameras to track the authors' bodies so that the movements can be analyzed and behaviors can be made automatically. The following examples illustrate the guideline:

- "The number of triangles on high polygon objects were reduced to optimize the cutting time to an order of magnitude of seconds" [37];
- "In other words, the interaction manager enables developers to create events that are easy to configure and are applied automatically to the characters" [38];
- "The idea is to provide users with a modeling tool that intuitively uses the reality scene as a modeling reference for derivative scene reconstruction with added interactive functionalities" [10].

### 3.3.3. Customization

This guideline refers to giving the author enough control over changes; application of 3D content anywhere in the virtual environment; alignment of 3D virtual models to scale, position and orientation; appearance configuration of 2D and 3D elements, changing color, size and shape; assigning behaviors and animations to a 3D mesh; assigning and combining functions and interactions between objects; modifying scene lighting, cameras, and environments; set timing, duration, start/end point, intensity, and direction of virtual

reality multisensory stimuli; create more expressive interactivity through action specification for VR hardware, such as controllers and/or haptic gloves; applying emotional state and personality to a virtual agent; specifying behaviors to react to discrete events such as user actions, system timers or collisions; customization of the virtual reality hardware while the software is always in an executable state; add annotations; edit texts; set pattern-specific parameters on software that uses them; organize the workspace layout by changing tools, tabs, and windows of the software. The following examples illustrate the guideline:

- "While the state-of-the-art immersive authoring tools allow users to define the behaviors of existing objects in the scene, they cannot dynamically operate on 3D objects, which means that users are not able to author scenes that can programmatically create or destroy objects, react to system events, or perform discrete actions" [8]—missing customization;
- "The system workflow design of VRFromX that enables creation of interactive VR scenes [...] establishing functionalities and logical connections among virtual contents" [10];
- "Some requests were [...] more freedom to change the parameters of the experience, i.e., to right click on 3D models and change the parameters of the assets on the fly" [41].

### 3.3.4. Democratization

This guideline relates to providing people with access to technical expertise via a radically simplified experience; authoring tools that can be accessed via a web browser (web-based) to make it easier for people to get started because they do not require the download of a special program and can be used for a variety of cases and purposes; web-browser application that can also be used through mobile devices that support virtual reality, bringing more accessibility and knowledge about virtual reality development; open-source and publicly available tools that can reach multiple researchers to build and evaluate them; the use of platforms such as GitHub to share resources and encourage users to contribute their own assets; empowerment of the citizen-developer model, with no-code procedures to design and develop virtual reality applications; hardware popularization with lower costs and better quality; the use of free stores for the distribution of applications and plug-ins at no cost; the use of libraries and frameworks such as Three.js and A-FRAME for web-browser development. The following examples illustrate the guideline:

- "[...] the advances of WebVR have also given rise to libraries and frameworks such as Three.js and A-FRAME, which enable developers to build VR scenes as web applications that can be loaded by web browsers" [8];
- "FlowMatic is open source and publicly available for other researchers to build on and evaluate" [8];
- "[...] democratization is focused on providing people with access to technical expertise (application development) via a radically simplified experience and without requiring extensive and costly training" [41].

### 3.3.5. Metaphors

This guideline refers to turning abstract concepts into tangible tools; use of visual resources and gestures to execute actions in the virtual world in a similar way to the real world, which improves the author's immersion; beginning actions with natural interactions and manipulation, for example, inserting a virtual disk into a virtual player as a start trigger to play music; move and position objects as if they were in the real world; use of buttons on the controllers to reproduce actions similar to what we would do in real life, such as pulling the trigger button to grab an item and releasing it to drop; use of miniatures to localize things at a glance on the interface; connection of objects distant from each other by making the physical movement of drawing visible lines between them; the use of visual icons, such as fire and ice, to represent haptic feedback, such as warm and cold, to the user, inducing a multisensory experience; the use of different shapes and colors to represent different types of data; the use of numbers to indicate sequences; the use of hologram overlays to show the

content of a pattern or object before interacting with it; real-world and virtual events that can be linked using IoT-enabled devices, for example, starting an object print in a virtual printer can start the process on a physical 3D printer; not using the correct metaphor can sometimes lead to a user misinterpreting the tool or action; in different contexts such as collaborative work, metaphors can naturally appear, such as the formation of individual territories when working in groups in the same space. The following examples illustrate the guideline:

- "They can draw edges to and from these abstract models to specify dependencies and behaviors (for example, to specify the dynamics of where it should appear in the scene when it shows up)" [8];
- "Similar to Alice in Wonderland, the users will gradually shrink as they trigger the entry procedure. Authors can access the world in miniature model and experience it in full scale to make changes to the content" [39];
- "Compared to the logic used in the construction of interactions, the task construction uses generic activities which should be also clear to novices without a technical background since they are comparable to actions in the real world" [12].

### 3.3.6. Movement Freedom

This guideline concerns using body movements to simplify creation and interaction while authoring in virtual reality immersion; use of 3D hand drawing (not only 2D) to retrieve 3D models, even with non-perfect sketches; freedom to explore the space, touch objects, manipulate elements, and encounter other users in a flexible virtual space; having the ability to move freely and safely in the virtual world, zooming in and out without needing to change positions in the physical world; immersive editing of programming elements through direct manipulation; manipulation of virtual objects using movements similar to those in the physical world, which can also be interpreted as a metaphor; free arrangement of elements anywhere in the virtual space; interacting with and editing 3D elements through simple hand gestures in a free-form manner, for example, by selecting an area of the 3D to be cut; creation of organic 3D shapes through immersive modeling; organization of a workspace using all the extensions of a virtual environment; having not only the option to work individually but also access another user by moving toward them to share items or communicate; having different options to visualize all the extensions of an element, either by rotating it, physically moving around it, or even going through it to have different points of view. The following examples illustrate the guideline:

- "One reason is that through direct manipulation users can feel more immersed—as if the wire is in their hands" [36];
- "A brush tool was developed which enables users to select regions on point cloud or sketch in mid-air in a free-form manner" [10];
- "Users can also perform simple hand gestures to grab and alter the position, orientation and scale of the virtual models based on their requirements" [10].

### 3.3.7. Optimization and Diversity Balance

This guideline relates to the reduction in steps to authoring experiences without limiting creative freedom, which can often be achieved with the application of other guidelines such as automation, visual programming, and reuse; giving the authors the feeling of completing more activities in less time, by reducing the number of inputs to obtain a result; reduction in ambiguity between views in 2D and 3D by authoring in immersion, so the user does not have to spend a lot of time imagining projections; improving the efficiency of the editing process through collaborative work with many users; use of a programming language that is easy to use and has free codes available from outside libraries; positioning of priority items physically close to the user, such as keeping a set of tools always attached to the author's hand; avoid complexity and unnecessary actions, which can lead to incomprehension, impatience, and fatigue for authors; not showing all training materials at once to reduce cognitive load; organization of functionalities in patterns and

categories to focus attention during development; the use of the right rendering modes and making the best use of the hardware to always obtain good graphics and performance; combination of simple elements to create others that are more complex. The following examples illustrate the guideline:

- "To make our system more efficient, we have to limit the capabilities of the Action entity targeting simple but commonly used tasks in training" [17];
- "The construction uses two dialogs to create the task and the activities so that the novice only needs to focus on the current task or activity" [12];
- "We decreased further the complexity by using wizards to focus the user on smaller steps in the development" [12].

The following sections describe the design guidelines classified as features.

### 3.3.8. Documentation and Tutorials

This guideline refers to educating the user while using a tool, demonstrating the step-by-step process in real time; using diversified resources to present how to execute a function, such as images, animations, recorded videos, text, audio guidance, holographic icons, and virtual embodied characters; creating specific initial tasks to teach basic tools on practice; publish tutorials in a variety of places, including YouTube, software documentation, and online forums; making sure to include missing information reported by users to complement the materials; encouraging online communities to create more knowledge about the tool; inclusion of error messages to help the user understand what not to do and how to recover activities; making sure that help buttons are visible and easily accessible; avoiding the presentation of too many steps at once, keeping enough details and a logical structure to follow; the use of automation to detect when the user is having difficulties to move on with a task and provide an insight to solve that. The following examples illustrate the guideline:

- "For each step, instructions are visualized as text in the menu to help participants remember which step they are performing" [37];
- "We believe that more visual aid in the form of animations showing the movement path can help ease the thinking process of participants" [37];
- "Documentation would be another interesting direction in the future, as two participants said they preferred A-FRAME in the sense that the APIs documentation was detailed and easy to understand" [8].

### 3.3.9. Immersive Authoring

This guideline relates to avoiding 2D-display or projections while creating a virtual world; performing multiple activities while immersed and use the immersion to improve the author's creation experience by, for example, executing a sketch in 3D to start a search for assets, 3D modeling, programming, building scenes or environments, reading documentation, and interacting with other authors; enjoying an immersive experience that has been deployed is not the same as creating this experience using virtual reality as a development tool, as the first option is only available to the final user; when applied with real-time feedback, immersive authoring creates a *what you see is what you get* (WYSWYG) experience; reducing the abstraction needed to convert 2D information to 3D; allowing users to share information and resources while they are in the same space and working with other people; a good immersive authoring interaction in virtual reality is heavily influenced by movement freedom and haptic feedback; to fit with the user view extension and avoid visual pollution or confusion, the immersive user interface must be simplified; avoidance of switching back and forth between the 2D and 3D screens to check how things are displayed in immersion; well applied to testing virtual reality functions in real-time and debugging; one issue is that wearing a head-mounted display for an extended period of time can be exhausting. The following examples illustrate the guideline:

- "[...] expedites the process of creating immersive multisensory content, with real-time calibration of the stimuli, creating a "what you see is what you get (WYSWYG)" experience" [35];
- "[...] immersive authoring tools can leverage our natural spatial reasoning capabilities" [36];
- "With the lack of additional spatial information and the disconnection between developing environments (2D displays) and testing environments (3D worlds), users have to mentally translate between 3D objects and their 2D projections and predict how their code will execute in VR" [8]—(missing immersive authoring).

### 3.3.10. Immersive Feedback

This guideline relates that, in virtual reality, action feedback is both visual and haptic/physical, using hardware parts, such as head-mounted displays, controllers, and wearable as an extra interaction source; immersive experience feedback can have multiple formats, from rendered icons and symbols to haptic stimuli such as controller vibrations. It is possible to accurately represent physical stimuli such as thermal, vibrotactile, and airflow, which require different hardware to reproduce and can be costly or inflexible, but would increase immersion [44]; users accept creative solutions such as animations, sounds, and icons representing physical stimuli as feedback; visual and haptic feedback must occur in real-time to be accurately felt, or the user will not engage with the application if the stimuli arrive at the wrong time; the tool must allow the author to apply immersive feedback and preview the results before releasing them; associating behaviors from virtual elements to the tracking of hardware (head-mounted displays and controllers); configuring controller buttons to start logical operations to facilitate the development, such as activating a virtual menu attached to the hand or a frequently used function. The following examples illustrate the guideline:

- "Rendering haptic feedback in virtual reality is a common approach to enhancing the immersion of virtual reality content" [33];
- "[...] various types of haptic feedback, such as thermal, vibrotactile, and airflow, are included; each was presented with a 2D iconic pattern. According to the type of haptic feedback, different properties, such as the intensity and frequency of the vibrotactile feedback, and the direction of the airflow feedback, are considered" [33];
- "The use of multisensory support is justified by the fact that the more the senses engaged in a VR application, the better and more effective is the experience" [35].

### 3.3.11. Real-Time Feedback

This guideline relates to a real-time visualization or physical perception of what is being authored, related either to 3D editing, code compilation, animation preview, or hardware set-up used for a scene; help avoid making mistakes while creating, as you do not need to wait until the end to see the result; minimizing latency [44]; it allows non-experts to spot mistakes much more quickly; visual representation of actions performed on objects, such as a wireframe highlight to describe the geometry selection and an animation preview to show if the behaviors attached to an object really work; view the editing actions of other users in collaborative sessions as they occur simultaneously; preview of multisensory physical stimuli, such as wind, heat, or vibration, while applying them to objects, despite the fact that they are frequently created through code in a 2D screen; available either for conventional 2D monitors or head-mounted display devices; allows better fine-tuning of the experience; when associated with immersive authoring, real-time feedback enables content creators to have a *what you see is what you get* experience, which means the user has a real view of the virtual environment while composing the scene; the authoring tool must allow the author to choose between turning on or off this feature as it can often cause issues and delay during the initialization of complex scenarios due to the quantity of information. The following examples illustrate the guideline:

- "AffordIt! offers an intuitive solution that allows a user to select a region of interest for the mesh cutter tool, assign an intrinsic behavior, and view an animation preview of their work" [37];
- "We believe that more visual aid in the form of animations showing the movement path can help ease the thinking process of participants" [37];
- "The novices are supported in the construction by visualizing the interactive VR scene in the development. This ensures direct feedback of added entities to the scene and modified representative parameters of the entities inside the scene. This enables the novice to spot mistakes immediately" [12].

### 3.3.12. Reutilization

This guideline concerns optimizing the development time by retrieving relevant elements from a collection or library, such as 2D/3D objects, audio files, codes to set behaviors and interactions, animations, lighting, and so on, so the author does not always need to have advanced knowledge in 3D modeling or programming [44]; libraries and collections must be integrated into the software so the user does not need to access external sources and go through different processes to import different formats of files to the authoring tool, facilitating scene creation; integrating popular libraries into the tool leads to a bigger variety of models, considering that more authors are collaborating with these libraries; it is difficult to find the right element in large libraries; automation processes, such as artificial intelligence networks trained to search for 3D assets in the virtual world using free-hand sketches, can help; saving author's creations for later is another form of reusing things; having templates helps start content creation in authoring tools; photogrammetry is an automated way to retrieve objects from the real world using cameras. The following examples illustrate the guideline:

- "We propose that by utilizing recent advances in virtual reality and by providing a guided experience, a user will more easily be able to retrieve relevant items from a collection of objects" [32];
- "[. . . ] we propose intuitive interaction mechanisms for controlling programming primitives, abstracting and re-using behaviors" [8];
- "Users can also save the abstraction in the toolbox for future use by pressing a button on the controller" [8].

### 3.3.13. Sharing and Collaboration

This guideline relates to the creation and manipulation of virtual space via collaborative works in which multiple and disparate stakeholders can use their imaginations while working with multisensory immersion from a local or remote network; following each other's activities in real-time; present ideas, products, and services to stakeholders, executives, or buyers in a business context; speeding up the creation process with more workers dealing with different tasks at once; enabling virtual round-tables for creative works, improving prototyping processes; combining the knowledge of different professionals in the same experience; edit of different objects at the same time by different users; people with more immersive technology experience can better assist and guide beginners while sharing the same space; users can change how others perceive them by customizing the color and shape of their avatars [44]; in sharing activities, tasks can be assigned and materials can be switched between users, such as 3D and 2D assets, text documents, textures, etc; create specific tools to enable a better experience in collaborative mode, such as setting mechanisms to lock the editing of an object by a user while it is being edited by another person; different groups of people will interact in different ways and at different levels and frequencies, changing the format of the discussions and also establishing social protocols such as owning objects and claiming territory in virtual space. The following examples illustrate the guideline:

- "[...] directly transmitted to others, and they can observe the doings of others in real time. The users work together on a virtual scene where they can add, remove, and update 3D models" [34];
- "This is useful because multisensory VR experiences might require multiple features that are produced by different professionals, and a collaborative feature will enable the entire team to work simultaneously" [35];
- "Each user is uniquely identified by a floating nameplate and avatar color. The same color is also used for shared brush selections. This allows users to see the actions of others to support collaborative tasks and information sharing, as well as to avoid physical collisions" [40].

### 3.3.14. Visual Programming

This guideline concerns programming through dataflow instead of creating text lines of code to create behaviors and reactions for the scene components, characters, and objects, which leads to a reduction in text inputs; the use of geometrical formats as nodes that already have a function applied to them, so the author does not need to re-write the text or even know how to do it; all the functions and connections can be presented in a graphic and optimized way; the Blueprint from Unreal Engine, applied through 2D interaction, is a well-known and well-implemented visual programming format in the world of game engines; there are already a variety of formats for the Visual Programming Languages, and they can be implemented in both desktop editing mode (2D screen) and immersive authoring mode (using head-mounted display); the nodes containing preset functions can be called *primitives*; the connection between primitives is also visual, being usually represented by edges going from one node to the other; still, this format can have problems becoming too complex when the codes get too big; it helps with reutilization as the abstraction of functions as groups of nodes can be saved, duplicated, and united to create more complex functions; it can speed up the process of prototyping behaviors. The following examples illustrate the guideline:

- "FlowMatic uses novel visual representations to allow these primitives to be represented directly in VR" [8];
- "Unreal Blueprint, a mainstream platform for developing 3D applications, also uses event graphs and function calls to assist novices in programming interactive behaviors related to system events" [8];
- "The development of a visual scripting system as an assistive tool aimed to visualize the VR training scenario in a convenient way, if possible fit everything into one window. The simplicity of this tool was carefully measured to provide tools used also from non-programmers. From the beginning of the project, one of the main design principles was to strategically abstract the software building blocks into basic elements" [17].

Figure 4 brings a visual depiction of the developed design guidelines.

### 3.3.15. Additional Considerations

Other factors were frequently mentioned in the reviewed works, such as engagement, fun-to-use, immersiveness, physical comfort, graphical quality, and the acquisition cost of equipment. These factors are not directly involved with intuitiveness, as the guidelines are, because they could be considered as consequences or even challenges related to the democratization of virtual reality technology. Therefore, there is a distance from the design guidelines proposal for the development of more intuitive virtual reality authoring tools, but they are related to the general structure of the technology in our society, aiming for a more popular use among people with various levels of knowledge.

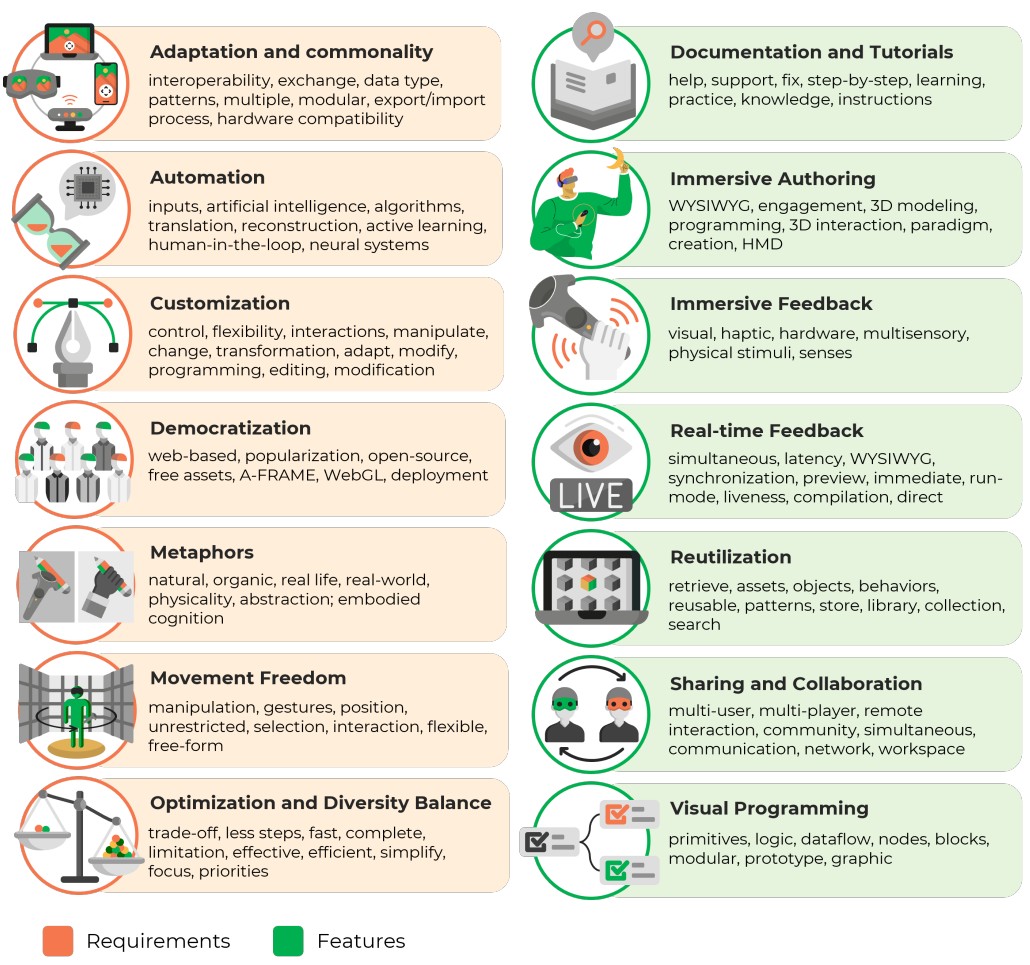

**Figure 4.** The fourteen developed design guidelines.

Concerning the term's definition, *engagement* stands for the active participation and involvement of the user with the tool, indicating that the users have a high level of engagement with the experience provided. As a complement to engagement, the term *fun-to-use* is used to describe the tools with which the users were not only focused on completing the tasks but also having a good time doing it, being frequently related to the concept of gamification. *Immersiveness* is the quality of an experience that provides the user with deep absorption and makes them feel like they have been transported to another world through multisensory feedback, not just based on images and sounds.

The lack of *physical comfort* is frequently mentioned since virtual reality equipment is heavy, meaning that spending a lot of time with it can be exhausting and cause motion sickness in some people. *Graphical quality* is another factor pointed out as being missing from virtual reality experiences due to the geometry optimization necessary to be processed by the head-mounted display. *Graphical quality* has a direct effect on how immersed you feel in an experience, since the better the graphics, the more immersed you feel. Finally, the high *cost* of good virtual reality equipment acquisition is pointed out as one of the main reasons why the technology has not been well popularized so far. The following are quotations from the reviewed works that used the terms:

- Engagement: "The system also provides an engaging and immersive experience inside VR through spatial and embodied interactions" [10];
- Fun-to-use: "[. . .] the results of the statement if the participant had fun constructing the interactive VR scene suggests that VREUD supplies novices with a playful construction

of interactive VR scenes, which could motivate them to develop their first interactive VR scene" [12];

- Immersiveness: "[. . . ] the majority of VR applications rely essentially on audio and video stimuli supported by desktop-like setups that do not allow to fully exploit all known VR benefits" [35];
- Physical comfort: "[. . . ] some participants commented that navigating the virtual world could cause slight motion sickness" [8]/"[. . . ] we could observe impatience and fatigue when the participants had to type in the text for the callouts using the immersive technology (a virtual keyboard) or had to connect the nuggets to bring them in chronological order" [39];
- Graphical quality: "One disadvantage of these tools is that they do not support highly photorealistic graphics and first person view edits which are achievable only by Unreal Engine and professional CAD software in runtime environments" [41];
- Cost: "It lowers the cost as the templates and the abstractions replace the Application Designer and Programmer by standardizing" [41]/"These wider application areas of VR require, besides affordable devices, a usable process of authoring to reach the full potential" [12].

Another key consideration is how, in practice, the design guidelines should positively contribute to the growth of the metaverse through their impact on the development of easier-to-use authoring tools and, consequently, the increase in the volume of virtual world creation.

The computing power and programming required to create virtual worlds and the accurate physical behavior of related objects are discussed [45]. In a metaverse architecture, the concept of *metaverse engine* is presented, which includes software technologies used in the creation of virtual worlds, such as immersive technologies (virtual reality, augmented reality, mixed reality), brain–computer interaction, artificial intelligence, digital twins (3D creation), and blockchain [23]. Ideally, the *metaverse engine* would use big data coming from the real world in an automatic way to create, maintain, and update the virtual world. The virtual economy would come from virtual avatars doing things on their own, such as trading personalized content made by AI to improve the metaverse ecology.

Contrasting these ideas, human developers are still in charge of making virtual worlds for the metaverse. Because of that, *virtual world engines* will become a standard feature of the metaverse, much like English is a standard language in the world, as the global economy continues to shift to virtual worlds [45]. Besides the many advantages presented by mainstream game engines such as Unreal and Unity, there is still a lot of discussion on what is the easiest and best way to build the metaverse, including how to facilitate the exchange of information, virtual goods, and currencies between these virtual worlds.

The *integrated virtual world platforms* (IVWPs) are a new approach to dealing with the creation of virtual worlds that "are designed so that no actual coding is required. Instead, games, experiences, and virtual worlds are built using graphical interfaces, symbols, and objectives [. . . ] The IVWPs interface enables users to create more easily and with fewer people, less investment, and less expertise and skill [45]".

This definition is very similar to those used to refer to authoring tools in several of the works analyzed here, but a difference can be seen by looking at the context in which each idea is used. While some work delves into game development, bringing examples such as Roblox, Minecraft, and Fortnite Creative [45], which are platforms that reach thousands of users and make thousands of dollars, authoring tools developed in the academic context are seen as proof-of-concepts with non-profit goals and are most often applied in the professional environment, not entertainment. Furthermore, it is interesting to see how both integrated virtual world platforms and authoring tools share not only concepts but also challenges, such as the fact that both "wants to enable creators' creative flexibility while standardizing underlying technologies, maximizing interconnectivity among everything that's built, and minimizing the need for training or programming knowledge on the part

of creators" [45]. Therefore, these platforms are more difficult to develop than the Game Engines mentioned above, as every feature becomes a priority.

Facilitating virtual reality development is also not a priority in the mainstream when it comes to integrated virtual world platforms, since one of today's biggest platforms, focused on virtual reality and augmented reality, Facebook's Horizon World remains small when compared to Roblox, which provides immersive virtual reality but prioritizes traditional screens [44].

As for which platform to use for the metaverse, due to the diversity of potential applications, the high technical level of difficulty to unite all of them in something unique, and given the speed at which new platforms are emerging, the best solution would be to handle all existing tooling options simultaneously, also avoiding market monopolization by a single corporation [45]. That is why gathering design guidelines could also affect the development of metaverses since it should help make authoring tools, or even integrated virtual world platforms, that are more intuitive for the people who make virtual worlds to use.

## 4. Conclusions

The fourteen gathered design guidelines can support the development of more intuitive authoring tools focused on authors with limited prior knowledge of 3D modeling and programming, resulting in a more sustainable virtual reality. With the evolution of immersive technologies, many of these guidelines are becoming easier to implement. However, it is also important to understand the intended audience and demand so the priority guidelines for that context can be defined. Even so, the main contribution of this research is the systematic organization and classification of widely used themes and concepts in virtual reality since none of them were invented in this research, creating, in other words, ontologies.

In this study, three research questions were satisfactorily addressed. To answer the *Q1: What are the characteristics of the virtual reality authoring tools reviewed?*, we extract important information related to the VR authoring tools developed in the reviewed articles, such as the artifact definition, software, and hardware tools used in the development process, as well as their plugin or standalone type classifications. In addition, we highlight important general characteristics of these tools, such as their ability to create virtual environments, incorporate 3D models, and serve a general purpose, as they can be used to create VR experiences for a variety of fields. To answer *Q2: What is the definition of intuitiveness in virtual reality authoring tools?*, we compiled key quotes from the reviewed articles that exemplified intuitiveness. As we have seen, it is not yet possible to evaluate or measure intuitiveness objectively, but other metrics such as usability, effectiveness, efficiency, and user satisfaction can be used to indicate a tool's intuitiveness. Such techniques were employed in the reviewed articles, indicating that the authoring tools developed were intuitive, according to our qualitative interpretation of the definition of intuitiveness. Thus, we hypothesized that by evaluating the characteristics of such intuitive tools, we would be able to respond to *Q3: What are the guidelines for designing intuitive virtual reality authoring tools?*, which was accomplished based on the fourteen design guidelines presented.

In practice, these guidelines can be used as a starting point for software developers during the project exploration phase, assisting them in defining the requirements and features of their virtual reality authoring tool. The guidelines can also be used to evaluate the intuitiveness of existing virtual reality authoring tools when applied to a research methodology. Other findings and contributions of this study included discussions about the lack of ontologies and taxonomies related to virtual reality authoring tools and how the guidelines can aid in the development of the metaverse. We discovered that the guidelines themselves may become ontologies and/or taxonomies, while the influence of the creation of more intuitive virtual reality authoring tools should increase the number of people capable of creating their own content to compose the virtual worlds and VR experiences of the metaverse since non-experts would also be able to use them.

As a limitation of this study, the design guidelines were derived from the reviewed articles, which means other guidelines may not be identified by our literature search. In addition, many of the guidelines are also connected to software development principles in general; therefore, some of them could be applied to applications not related to virtual reality. Moreover, our categorization of the guidelines is subjective; they could be organized into different categories.

Concerning future research, each guideline provides the opportunity to delve deeper into the definition of a technical software development approach, even to the point of creating a subclass. Since the guidelines are never presented in isolation, it would be interesting to analyze the correlation level between them, to better understand how to apply them in a project. In addition, it is necessary to conduct actual tests in the context of application development using the guidelines in order to comprehend their impact on the project definition. Finally, it is possible to test the use of the guidelines as a system for evaluating existing authoring tools from the perspective of novice professionals or individuals with no background in 3D modeling or programming.

Above all, this research focused on the democratization of tools for creating virtual worlds, which directly impacts the faster and more sustainable advance and dissemination of trends such as the metaverse. Because of this, virtual reality technology will keep helping to reach the UN Sustainable Development Goals by giving more people the chance and independence to create immersive experiences and develop skills for the digital transformations of society.

**Supplementary Materials:** The following are available online at https://www.mdpi.com/article/10.3390/su15042924/s1.

**Author Contributions:** Conceptualization, I.L.C. and I.W.; methodology, I.L.C. and I.W.; validation, I.W., A.L.A.J., T.B.M. and C.V.F.; formal analysis, I.W. and A.L.A.J.; investigation, I.L.C.; resources, I.W., A.L.A.J. and T.B.M.; data curation, I.L.C.; writing—original draft preparation, I.L.C.; writing—review and editing, I.W., A.L.A.J., T.B.M. and C.V.F.; visualization, I.L.C.; supervision, I.W., A.L.A.J. and T.B.M.; project administration, I.L.C. and I.W. All authors have read and agreed to the published version of the manuscript.

**Funding:** This research was funded by the Virtual and Augmented Reality for Industrial Innovation Lab at SENAI CIMATEC University Center.

**Institutional Review Board Statement:** Not applicable.

**Informed Consent Statement:** Not applicable.

**Data Availability Statement:** The data presented in this study is fully available in the Supplementary Material.

**Acknowledgments:** The authors would like to thank the financial support from the National Council for Scientific and Technological Development (CNPq). Ingrid Winkler is a CNPq technological development fellow (Proc. 308783/2020-4). We also gratefully acknowledge Johann H. S. Erickson's insightful comments.

**Conflicts of Interest:** The authors declare no conflict of interest.

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
