# Peer review of "Towards Sustainable Virtual Reality: Gathering Design Guidelines for Intuitive Authoring Tools"

_sustainability, doi:10.3390/su15042924_

Round 1

Reviewer 1 Report

In this survey paper, the authors perform a literature survey in VR to evaluate their design guidelines' efficacy and suggest effective authoring tools. While the reviewer commends the authors' effort for their exhaustive study of the literature to come up with such detailed findings, the reviewer believes that this paper has numerous flaws.

  1. There are several issues with grammar and sentence structures in the paper.
  2. Figure 1 needs to be in high resolution,
  3. How do the authors extract the frequent terms in table 3? Do they estimate manually or utilize some NLP pipeline for this?
  4. The majority of the paper's body is driven by lengthy discussions and the author's discussions which don't have any metrics to validate except the statements from the existing literature. The authors do not present any quantitative analysis, such as statistics or numeric metrics, to validate the arguments in the paper. Most of the discussion is explanatory arguments that the authors align with some cited statements from the surveyed literature.
  5. Although the authors make some critical claims regarding the flaws in the existing literature at the beginning of the manuscript, it needs to be clearly discussed how the authors hypothesize solutions about these aspects and provide some statistical results to validate their arguments.

The reviewer believes this paper adds minimal value to the literature and doesn't have strong takeaways for its readers. For this reason, the reviewer doesn't recommend the paper for publication.

Author Response

Dear Reviewer,

We appreciate your effort reviewing our manuscript, Towards sustainable virtual reality: gathering design guidelines for intuitive authoring tools. We revised it according to your recommendations, which significantly contributed to the improvement of the work.

Please find below a detailed description of the adjustments made:

Point 1 - There are several issues with grammar and sentence structures in the paper. 
Response 1 – Following the editor’s recommendation, we had our manuscript carefully checked by a native English-speaking colleague. The following were modified, among others: the use of contractions, acronyms, prepositions, conjunctions, the verbal conjugation, punctuation of the guidelines description, occurrences in which the cited work appeared as the subject of a phrase, and the reading flow of the entire document.

Point 2 - Figure 1 needs to be in high resolution 
Response 2 – Thank you for the suggestion, we updated the image with a better resolution.

Point 3 - How do the authors extract the frequent terms in table 3? Do they estimate manually or utilize some NLP pipeline for this?
Response 3 – We appreciate your suggestion. Because this is exploratory research using a qualitative approach, we changed the expression "frequent terms" to "related terms" (line 235 and Table 3 caption). Additionally, to make clearer the qualitative design of our work, we also revised the manuscript and provided the following information about the data interpretation and coding process: 
(1) we specified that text non-numerical analysis was used for the data analysis procedures for interpreting and validating collected data and that the type of interpretation consisted of themes and patterns interpretation to identify design guidelines  (lines 201-207);
(2) we included a detailed description of the hand code data process followed (lines 208-228), as well a detailed description of how the terms were extracted (lines 234-239);

Point 4 - The majority of the paper's body is driven by lengthy discussions and the author's discussions which don't have any metrics to validate except the statements from the existing literature. The authors do not present any quantitative analysis, such as statistics or numeric metrics, to validate the arguments in the paper. Most of the discussion is explanatory arguments that the authors align with some cited statements from the surveyed literature.
Response 4 – Our work is exploratory study adopting a qualitative approach, which is appropriate given the lack of prior research on the development of intuitive authoring tools for virtual reality. Our criteria for choosing a qualitative design is that the problem does not involve the identification of factors that influence an outcome, the utility of an intervention, or the identification of the most accurate predictors of outcomes, which would be best addressed with quantitative methods. In contrast, a qualitative approach is required if a concept or phenomena requires exploration and comprehension because there has been little research on it or because it involves an understudied sample [1]. 
Thank you for bringing this to our attention, to clarify that our study employed a qualitative design, we revised the manuscript and provided the following information: 
(1) we stated that the study adopts qualitative approach and justified the motivations for this choice (lines 110-115); 
(2) we clarified that in qualitative research design the type of data analyzed is text information, which allow researchers to interpret themes or patterns that emerge from the data (lines 200-203); 
(3) we specified that text non-numerical analysis was used for the data analysis procedures for interpreting and validating collected data and that the type of interpretation consisted of themes and patterns interpretation to identify design guidelines  (lines 203-206);
(4) we included a detailed description of the code data process followed (lines 207-232) and we changed the expression "frequent terms" to "related terms" (line 235 and Table 3 caption);
(5) we clarified that in terms of qualitative validity, we adopted the peer debriefing strategy to assess the validity of the identified design guidelines (lines 244-252).
(6) because many qualitative studies include visuals, figures, or tables as adjuncts to the discussions [1], we generated the Figure 4, which is a visual depiction of the developed design guidelines.
(7) we created a flowchart (Figure 2) summarizing the process described in 2.5. Synthesizing and Analyzing section;
(8) Qualitative research is interpretive research [1], so  we clarified that the authors have a strong background and experience with 3D visualization tools, graphical design, computer-aided design, and software such as SolidWorks, Adobe Photshop, CATIA, and Autodesk, among others (lines 125-129).

Point 5 - Although the authors make some critical claims regarding the flaws in the existing literature at the beginning of the manuscript, it needs to be clearly discussed how the authors hypothesize solutions about these aspects and provide some statistical results to validate their arguments.
Response 5 - In a qualitative study, inquirers state research questions, not hypotheses [1]. Our research questions Q1, Q2 and Q3 (lines 142-144) adhered to the recommendation that they be broad questions that ask for an exploration of the central phenomenon or concept in a study. Rather than employing statistical analysis and statistical interpretation like quantitative research, qualitative research employs non-numeric text as data analysis procedures, and themes and patterns interpretation in order to generate hypotheses and further investigate and understand quantitative data [1]. We improved this topic including a summary of the main findings answering our research questions Q1, Q2 and Q3 (lines 638-639) and we also presented how our findings can be put to practical use (lines 640-641).

[1] Creswell, J. W., & Creswell, J. D. (2017). Research design: Qualitative, quantitative, and mixed methods approaches. Sage publications.

Reviewer 2 Report

The paper is well written and and the design flow is well formulated. I just have few concerns, highlighted as:

1. "Ref" must be named with their specific authors

2. Can the authors incorporate a flowchart for their proposed method towards design guidelines.

3. Please justify why references are used for some sub-headings. I suggest the authors can incorporate their own categorisation.

Author Response

Dear Reviewer,

We appreciate your effort reviewing our manuscript, Towards sustainable virtual reality: gathering design guidelines for intuitive authoring tools. We revised it according to your recommendations, which significantly contributed to the improvement of the work.

Please find below a detailed description of the adjustments made:

Point 1 - "Ref" must be named with their specific authors

Response 1 - We revised the whole manuscript and fixed any instances where the referenced work served as the subject of a sentence. We also had our manuscript carefully checked by a native English-speaking colleague and the following were modified, among others: the use of contractions, acronyms, prepositions, conjunctions, the verbal conjugation, punctuation of the guidelines description, and the reading flow of the entire document.

Point 2 - Can the authors incorporate a flowchart for their proposed method towards design guidelines.

Response 2 – Thank you for the suggestion, we created a flowchart (Figure 2) summarizing the process described in 2.5. Synthesizing and Analyzing section. The previous method steps were described in Figure 1, containing a systematic review flow diagram, adapted from PRISMA protocol. Additionally, to clarify the research qualitative design we provided the following information: 

(1) we clarified that in qualitative research design the type of data analyzed is text information, which allow researchers to interpret themes or patterns that emerge from the data (lines 200-203); 

(2) we specified that text non-numerical analysis was used for the data analysis procedures for interpreting and validating collected data and that the type of interpretation consisted of themes and patterns interpretation to identify design guidelines  (lines 203-206);

(3) we included a detailed description of the code data process followed (lines 207-232) and we changed the expression "frequent terms" to "related terms" (line 235 and Table 3 caption);

(4) we clarified that in terms of qualitative validity, we adopted the peer debriefing strategy to assess the validity of the identified design guidelines (lines 244-252).

(5) because many qualitative studies include visuals, figures, or tables as adjuncts to the discussions [1], we generated the Figure 4, which is a visual depiction of the developed design guidelines.

Point 3 - Please justify why references are used for some sub-headings. I suggest the authors can incorporate  their own categorisation.

Response 3 – We created all of the categorizations and design guidelines; to make it clearer, we deleted the references from the sub-headings mentioned.

[1] Creswell, J. W., & Creswell, J. D. (2017). Research design: Qualitative, quantitative, and mixed methods approaches. Sage publications.

Reviewer 3 Report

This paper is a valuable contribution to the field of VR, specifically in relation to the development of accessible authoring tools. The rationale for the study is well-argued and supported by appropriate literature. You have used appropriate and recognised methods to carry out a systematic review and to analyse the findings of that review. The findings are discussed well, although the discussion could be organised and presented more effectively rather than the "list under paragraph headings" approach currently used. 

The following items need to be dealt with specifically:

Please adjust the reference style in accordance with the journal style. Where references are referred to by name in the text, then the name and date should be used, rather than just the number in parenthesis. For example " ... this was recognised by Jones and Smith 2019 [22] ..." Please look back at articles in the journal to see the correct style.

It would be helpful to include some form of prioritisation or weighting in the discussion of the guidelines extracted from the literature. At the moment, the results consist of lists of topics under each guideline heading with little differentiation regarding the usefulness and/or importance of the guideline and its constituent parts. The analysis of the chosen literature could include an assessment of importance from, say, number of times mentioned in the articles, how long the topic was discussed in the articles, the importance given to the topic by the authors of the articles, and so on. 

The Conclusions section needs to be extended, to include a summary of the main findings and a short commentary on how each of the research questions was answered in the paper. At the moment the conclusions are rather thin - it would be particularly helpful for the authors to recommend not just further research, but also how their findings in this paper might be taken forward and be put to practical use.

In general the English is very good, but there are some places where phraseology could be improved, e.g. "This guideline concerns to ..." should simply be "This guideline concerns ..."

Author Response

Dear Reviewer,

We appreciate your effort reviewing our manuscript, Towards sustainable virtual reality: gathering design guidelines for intuitive authoring tools. We revised it according to your recommendations, which significantly contributed to the improvement of the work.

Please find below a detailed description of the adjustments made:

Point 1 - The findings are discussed well, although the discussion could be organised and presented more effectively rather than the "list under paragraph headings" approach currently used. 

Response 1 - In accordance with qualitative approach, we discussed each identified design guideline in depth in a qualitative narrative, along with quotations from the reviewed works from which we interpreted that guideline, to support the categorized themes. Although we recognized numerous quotations for each identified design guideline, we highlighted just three quotations as illustrative instances of each design guideline for the purpose of this article writing flow; the entire list of quotations is accessible as Supplementary Materials. 

We appreciate you raising this point; so, because many qualitative studies include visuals, figures, or tables as adjuncts to the discussions [1], we generated Figure 4, which is a visual depiction of the developed design guidelines.

Point 2 - Please adjust the reference style in accordance with the journal style. Where references are referred to by name in the text, then the name and date should be used, rather than just the number in parenthesis. For example " ... this was recognised by Jones and Smith 2019 [22] ..." Please look back at articles in the journal to see the correct style. 

Response 2 - We revised the whole manuscript and fixed any instances where the referenced work served as the subject of a sentence.  

Point 3 - It would be helpful to include some form of prioritisation or weighting in the discussion of the guidelines extracted from the literature. At the moment, the results consist of lists of topics under each guideline heading with little differentiation regarding the usefulness and/or importance of the guideline and its constituent parts. The analysis of the chosen literature could include an assessment of importance from, say, number of times mentioned in the articles, how long the topic was discussed in the articles, the importance given to the topic by the authors of the articles, and so on. 

Response 3 – Our work is exploratory study adopting a qualitative approach, which is appropriate given the lack of prior research on the development of intuitive authoring tools for virtual reality. Rather than employing statistical analysis and statistical interpretation like quantitative research, qualitative research employs non-numeric text as data analysis procedures, and themes and patterns interpretation in order to generate hypotheses and further investigate and understand quantitative data [1]. Thus, we intended to give equal status to all guidelines in this exploratory qualitative study because they are interpretations and agglutinations of what the authors of the reviewed studies claimed, and only a few of them explicitly quote any of the guidelines. An outcome showing how many times they were mentioned or how much the topic was discussed in the authors' articles would be fruitless, as we would acquire erroneous quantitative values. Hence, in the Conclusion section,  we propose that future research carry out a study of each guideline separately, which may include statistical estimations of each one's weight of relevance. 

Point 4 - The Conclusions section needs to be extended, to include a summary of the main findings and a short commentary on how each of the research questions was answered in the paper. At the moment the conclusions are rather thin - it would be particularly helpful for the authors to recommend not just further research, but also how their findings in this paper might be taken forward and be put to practical use.

Response 4 – Thank you for your contribution. We included a summary of the main findings answering our three research questions Q1, Q2 and Q3 (lines 638-639). We also presented how our findings can be put to practical use (lines 640-641).

Point 5 - In general the English is very good, but there are some places where phraseology could be improved, e.g. "This guideline concerns to ..." should simply be "This guideline concerns ..."

Response 5 – We corrected all instances of “This guideline concerns to [...]”. Additionally, we had our manuscript carefully checked by a native English-speaking colleague and the following were modified, among others: the use of contractions, acronyms, prepositions, conjunctions, the verbal conjugation, punctuation of the guidelines description, occurrences in which the cited work appeared as the subject of a phrase, and the reading flow of the entire document.

[1] Creswell, J. W., & Creswell, J. D. (2017). Research design: Qualitative, quantitative, and mixed methods approaches. Sage publications.

Sincerely,

Iolanda Lima Chamusca

Dr. Thiago B. Murari

Dr. Cristiano Vasconcellos Ferreira

Dr. Antonio Lopes Apolinario Junior

Dr. Ingrid Winkler

Round 2

Reviewer 1 Report

The reviewer appreciates the authors' effort to clarify the concerns and address them sufficiently. Commending the authors' effort, the reviewer would like to recommend the article for publication.